# Additively-Manufactured High-Concentration Nanocellulose Composites: Structure and Mechanical Properties

**DOI:** 10.3390/polym15030669

**Published:** 2023-01-28

**Authors:** Muhammad Latif, Yangxiaozhe Jiang, Jongmin Song, Jaehwan Kim

**Affiliations:** Creative Research Center for Nanocellulose Future Composites, Department of Mechanical Engineering, Inha University, 100, Inha-ro, Michuhol-gu, Incheon 22212, Republic of Korea

**Keywords:** nanocellulose, 3D printing, layered structures, polymer matrix composites, mechanical properties

## Abstract

Additive manufacturing technology (AMT) has transformed polymer composites’ manufacturing process with its exceptional ability to construct complex products with unique materials, functions, and structures. Besides limiting studies of manufacturing arbitrarily shaped composites using AMT, printed structures with a high concentration of nanocellulose face adhesion issues upon drying, resulting in shape fidelity issues and low mechanical strength. This research demonstrates an economical approach to printing a high-concentration (25.46 wt%) nanocellulose (NC) layer-wise pattern to fabricate structures. Two different composites are fabricated: (1) 3D-printed pure and high-concentration (10, 15, and 20 wt%) polyvinyl-alcohol (PVA)-blended NC structures followed by freeze-drying and impregnation of Epofix resin by varying hardener contents; (2) 3D-printed PVA-blended NC green composites dried at cleanroom conditions (Relative humidity 45%; Temperature 25 °C). Different contents (10, 15, and 20 wt%) of PVA as a crosslinker were blended with NC to assist the printed layers’ adhesions. An optimum PVA content of 15 wt% and an Epofix resin with 4 wt% hardener cases showed the highest bending strength of 55.41 ± 3.63 MPa and elastic modulus of 4.25 ± 0.37 GPa. In contrast, the 15 wt% PVA-blended NC cleanroom-dried green composites without resin infusion showed bending strength and elastic modulus of 94.78 ± 3.18 MPa and 9.00 ± 0.27 GPa, reflecting high interface adhesions as confirmed by scanning electron microscope. This study demonstrated that AMT-based nanocellulose composites could be scaled up for commercial use.

## 1. Introduction

Natural fiber-reinforced composites (NFRC) have recently gained much consideration to reduce our dependency on non-renewable reinforcements due to their superior mechanical properties, lightweight, low cost, and environment-friendly nature [1,2,3]. Natural fibers are categorized depending on their extracted sources: minerals, animals, and plants [4]. Plant fibers include core, leaf, grass, seed, bast, root, and wood [4]. Depending on the sources of natural fibers, NFRC is employed in different advanced applications ranging from automobile to airplane industries [5]. Among different natural fibers, nanocellulose, the most copious organic raw material on earth, has proved promising, offering innovative solutions to numerous critical contemporary societal challenges [6]. Cellulose, a polysaccharide produced mainly by plants, consists of linear glucan chains packed into microfibrils and held together through intramolecular hydrogen bonding and intermolecular van der Waals forces. These linear glucan chains are joined together by β-1, 4-glycosidic bonds, and cellobiose residues serve as the repeating unit at various degrees of polymerization. Cellulose is classed into three primary forms: bacterial nanocellulose, cellulose nanocrystals (CNC), and cellulose nanofibers (CNF) [7]. CNC and CNF are collectively referred to as nanocellulose (NC). NC materials have attracted increasing interest due to their alluring and exceptional qualities, including their abundance, high aspect ratio, superior mechanical strength, biodegradability, and biocompatibility. The copious hydroxyl (-OH) functional groups expanded NC applications in biosensors and actuators, biomedical devices, food packaging, nano transparent films for UV protection, triboelectric nanogenerators, and, as a reinforcement, in the form of woven fabric for composites [8]. So far, NC is mainly utilized as a nanofiller in composite materials, enhancing crystallinity, thermal, and mechanical properties. The main challenge of nanocellulose as a nanofiller is its homogeneous dispersion in a polymer resin to enhance the resulting composites’ functional properties [9].

Recently, numerous studies have been conducted to demonstrate the positive effect of nanocellulose nanofillers on the functional properties of polymer resin-infused composites [10,11,12]. Moreover, preparing continuous cellulose fibers with nanocellulose suspensions via either dry or wet electrospinning technology opens a wide range of possibilities for manufacturing NFRC [13]. Electrospinning is a valuable method to make nanocellulose long fibers (NCLFs) with diameters ranging from several microns to nanometers due to the action of electrostatic forces [14]. Numerous research studies have been published on the fabrication of NCLFs by either wet or dry electrospinning technology [15,16]. With the rapid development of fiber fabrication technologies, including wet electrospinning, several toxic aqueous solvents, such as sulfuric acid, hydrochloric acid, ethanol, calcium chloride, and acetone, were used for coagulation purposes of nanocellulose fibers which can be harmful to health and the environment [17]. Moreover, the high setup cost for electrospinning nanocellulose suspensions equipment and processing NLCFs to fabrics via traditional weaving setups is another challenging issue due to the nanosized diameter of filaments, reflecting expensive NLCFs compared to synthetic fibers [18,19]. High setup cost is the main reason for using nanocellulose as a nanofiller in polymer resins or as reinforced films in polymer composites. Due to this, Roneisha et al. [20] added CNC (nanofiller) in epoxy resin to increase the mechanical and thermal properties of the manufactured composites.

Similarly, Tuukka et al. [21] infused bio-epoxy in CNF aerogels to manufacture composites and studied their orientation and mechanical properties. Mehdi et al. [22] impregnated polymer to cellulose nanofiber networks to enhance the mechanical strength of the manufactured composites. Sandeep et al. [23] manufactured transparent CNF film composites using transparent epoxy resin. Hence, an economic system, such as additive manufacturing (3D printing), is essential for processing nanocellulose suspensions to NLCFs to save time, cost, and processing.

AMT’s advent opens a new way for engineering nanocellulose-reinforced polymer composites (NRPC) to create products with escalating complexity and improved functional performance [24,25,26]. 3D printing, also known as AMT, is an emerging processing method of fabricating complex shapes in a layer-wise pattern using computer-aided designs for various applications. The most significant aspects of 3D printing are removing restrictions in designing and constructing complex geometries and reducing material wastage [27,28,29]. Based on the ASTM 52900:2015 standard, AMT has seven classifications: direct energy deposition, material jetting and binder jetting technology, fused deposition modeling, vat photopolymerization, and material extrusion method [30]. Among different 3D printing technologies, material extrusion AMT is well-known for nanocellulose materials [31]. Material extrusion AMT can print nanocellulose in a layer-wise pattern to any complex shape structures at ambient conditions, enhancing the alignment of nanocellulose in the printing direction due to the use of micro-size diameter printing needles. These printing layers act as long filament-like electrospun fibers. Such economic technology can reduce the cost and time for manufacturing nanocellulose fabrics, opening a new way to fabricate inexpensive nanocellulose composites compared to synthetic fiber composites. The composite materials are not the latest, but limited studies are available in AMT [32]. Moreover, as far as we know, no study demonstrates a high concentration of nanocellulose impregnation Epofix and polyvinyl alcohol (PVA) to manufacture composites using AMT.

Contemplating the prevailing challenges, we proposed a feasible and efficient 3D printing technology to print a high concentration of nanocellulose (CNC/CNF) in a layer-wise pattern similar to electrospun fibers. The twin-screw extrusion was employed to eject the nanocellulose paste (25.45%) for 3D printing. Two different types of composites were manufactured: (1) 3D-printed pure and PVA-blended NC structures followed by freeze-drying and infusion of Epofix resin; (2) 3D-printed high-concentration nanocellulose (25.45 wt%) blended with high-concentration (15 and 20 wt%) PVA composites followed by cleanroom-drying (Relative humidity of 45%, and temperature of 25 °C) without an infusion of Epofix resin. The main reason for blending high concentrations (10, 15, and 20 wt%) of polyvinyl alcohol (PVA) with a high concentration of nanocellulose (25.45 wt%) is to improve the adhesion between printed layers because concentrated nanocellulose resulted in low bilayers adhesion. The impregnation of resin with varying hardener ratios (3 and 4 wt%) was performed via vacuum-assisted resin transfer molding (VARTM). The direct drying of the nanocellulose blended PVA composites in the cleanroom conditions was aimed at slowly drying so that the rapid shrinkage of the printed structures upon drying was prohibited with enhanced printed layers adhesion. Mechanical properties and cross-sectional morphology of all manufactured composites were then investigated via a three-point bending (TPB) test and scanning electron microscope (SEM). A schematic of the nanocellulose 3D printing process, followed by freeze-drying and resin infusion, is shown in Figure 1.

## 2. Experimental Section

### 2.1. Materials

Hardwood pulp, a mixture of Poplar and Aspen containing alpha-cellulose content of 85.7%, was bought from Chungnam National University, Daejeon, South Korea. The low molecular weight (22,000) PVA was obtained from Daejung Chemicals, Busan, South Korea. CNC was provided by CelluForce, Montreal, QC, Canada. Epoxy resin, Epofix, and its hardener triethylenetetramine (TETA) were obtained from Struers, Copenhagen, Denmark. The mold release agent Loctite frekote 700-NC was purchased from Henkel, South Korea.

### 2.2. Preparation of High NC Concentration Paste

The isolation of cellulose nanofibers (CNF) from hardwood pulp was accomplished via TEMPO-oxidation treatment followed by mechanical method (aqueous counter collision) [33]. The pure NC paste was obtained by mixing a high concentration of CNC and CNF (1.5~2 wt% relative to water) (CNF: CNC = 1:20) in a swing planetary mixer (SPM, FDU-2200, Tokyo Rikakikai Co. Ltd., Tokyo, Japan) at 1200 rpm for 5 min. Furthermore, the PVA-blended high-concentration NC paste was prepared by blending PVA (10, 15, and 20 wt% contents relative to water) with NC mixture in the SPM for 5 min at 1200 rpm. The compositions of prepared pastes are shown in Appendix A. The prepared paste (Appendix A) contains the highest possible NC (CNF and CNC) content. The primary aim of blending PVA in nanocellulose paste is to enhance the bonding/linkage between the layers, later confirmed by SEM and FTIR characterizations. 

### 2.3. Research Method

The current research process is divided into four stages to demonstrate the effectiveness of AMT in NRPC: paste preparation, printing process, drying process, and infusion process. Figure 1 shows the 3D printing setup. The NC paste (25.45 wt%) was filled into the 60 ml syringe and injected with a feed rate of 1.2 mL/min into the twin-screw extruder (MC 15 HT, Xplore, Sittard, The Netherlands) via a high-pressure syringe pump (Nexus 6000, Chemyx, Stafford, TX, USA). The homemade 3D printer connected with a nozzle of 1.51 mm diameter was directly attached to the twin-screw extruder through a 10 mm plastic pipe. The twin-screw extruder and printing speed of 9.37 mm/s and 150 rpm were optimized for high-concentration nanocellulose paste. A commercially available Cura software converted CAD files into G-code files. Based on the G-code, the first NC layers are compressed by 0.06 mm by needle and 0.20 mm for subsequent layers in the 3D printing process to enhance the adhesion of the layers. The printing parameters for high-concentration NC paste via a twin-screw extruder are depicted in Appendix A. The printability of the pastes (10, 15, and 20 wt% PVA-blended NC pastes) was confirmed by extruding them with the twin-screw extruder and making layers in a layer-wise pattern for manufacturing composites (Appendix A).

The 3D-printed structures were freeze-dried for 5 h to preserve the shapes with minimal shrinkage using a freeze-dryer (FDU-2200, Eyela, Tokyo, Japan). Since the freeze-dried structures are porous and fragile, a resin was infused via the VARTM method by varying hardener TETA contents from 3 to 4 wt%. For VARTM, the aluminum mold was sprayed with a mold-releasing agent (700 NCA) to ease the peeling off the structures. The freeze-dried structures were kept on the mold and covered with nylon cloth, followed by a resin flow net and vacuum bag. After the VARTM setup, the vacuum pressure (~0.4 MPa) was applied to remove air from the vacuum bag, followed by resin infusion. After resin infusion, the structures were kept at 60 °C for 6 hours. The VARTM procedure digital images are shown in Appendix A. Furthermore, the 3D structures were printed with optimized paste (15PVA-NC) and infused with Epofix resin to further ensure the proposed approach’s applicability.

Moreover, the green composites with 15 and 20 wt% PVA-blended high-concentration NC paste were 3D-printed and dried at cleanroom conditions (relative humidity: 45% and temperature: 25 °C) for 5 days. The printed composites were then further evaluated with mechanical and structural characterizations.

### 2.4. Mechanical Properties

Rectangular-shaped samples were printed for the TPB test as per ASTM D790 standards [34]. The TPB specimens were examined in a universal testing machine (TEST ONE, Busan, South Korea) to obtain flexural strength, modulus, and strain-at-break. For Epofix resin-impregnated samples, a size of 70 × 12.5 × 4 mm^3^ with a span length of 50 mm was kept during TPB. However, for PVA-blended NC cleanroom-dried samples, a size of 65 × 7 × 1.8 mm^3^ and a span length of 30 mm were maintained during the TPB test.

### 2.5. Structural Characterization

The chemical interactions and bilayers adhesion between components of the printed structures were examined by Fourier transform infrared (FTIR) spectroscopy (Cary 630, Agilent Technologies, USA) and scanning electron microscopy (SEM, S4000, Hitachi, Tokyo, Japan), respectively.

## 3. Results and Discussion

### 3.1. 3D Printing of Pristine, PVA-Blended NC, and PVA-NC Resin-Infused Composites

Figure 2 shows freeze-dried TPB specimens printed with the pure NC paste and the 15 wt% PVA-blended NC paste. As shown in Figure 2a, the pristine NC paste specimen showed cracks and poor adhesion between the printed layers, as mentioned by previous studies [35,36]. The 15 wt% PVA-blended NC paste samples after infusion are shown in Figure 2b. We observed improved adhesion between the printed layers when we used PVA as a secondary binder. It may be related to improved crosslinking due to hydrogen bonding at the printed layer interfaces. Figure 2c shows a 3D-printed wet box with length, width, and height of 21.51, 19.63, and 19.79 mm, respectively, printed with 15 wt% PVA-blended NC paste (15PVA-NC). The boxes showed negligible shrinkage without crack and adhesion issues after freeze-drying, as shown in Figure 2d, which might be due to the secondary binder (PVA) effect.

Box, cylindrical, and flower-shaped 3D structures were also printed with the 15PVA-NC to elaborate the suggested approach’s applicability, and the shapes are shown in Figure 3a,c,e. All printed shapes were freeze-dried and infused with the EP4 resin, as shown in Figure 3b,d,f. As expected, the final dried samples kept the prescribed shapes according to the designed CAD models. The suggested approach can easily print complex shapes with a high concentration of nanocellulose followed by resin infusion to make high-strength 3D structures compared to previously reported traditional 3D structures manufacturing methods.

### 3.2. Mechanical Properties 

#### 3.2.1. PVA-Blended NC-Resin-Infused Composites

The freeze-dried 3D structures were so porous and fragile that they were infused with resin. Epofix resin was impregnated to fill the pores of the freeze-dried structures via the VARTM process. A hardener (TETA) for the Epofix with different contents (3 and 4 wt%) was tested to maximize the 3D-printed structures’ mechanical properties. The Epofix resin with different TETA hardener content (3 and 4 wt%) was named EP3 and EP4. 

The TPB specimens were 3D-printed with pure and 10, 15, and 20 wt% PVA-blended NC pastes and freeze-dried to preserve the shape with minimal shrinkage. Then, they were infused with the Epofix resins (EP3 and EP4) via the VARTM process (Appendix A). The TPB specimens were evaluated in bending mode, and the results are shown in Table 1. Figure 4 shows the flexural strength and flexural modulus of all combinations of the pristine PVA-blended NC pastes infused with Epofix resins. The flexural strength, flexural modulus, and strain-at-break of the pristine NC paste composites increased when the hardener content was 4 wt% (EP4), attributed to the high degree of crosslinking between the resin and hardener, as shown in Figure 4. The stoichiometric relation between resins and their hardeners remarkably affects the mechanical properties [37]. The stress–strain curves for all combinations of pure and PVA-blended NC composites are shown in Appendix A.

The pristine high-concentration NC composites with EP4 resin showed a bending strength of 30.82 MPa, a bending modulus of 2.67 MPa, and a strain-at-break of 1.23%. However, EP4-infused composites showed a bending strength, modulus, and strain-at-break of 36.25 MPa, 3.19 GPa, and 1.19, respectively. The strain-at-break slightly decreased due to the composites’ brittleness at a high hardener ratio, resulting in improved crosslinking. In the cases of 10 wt% PVA-blended NC composites, the specimens showed the best flexural strength of 49.18 MPa, flexural modulus of 3.99 GPa, and strain-at-break of 1.312 for the EP4 resin (10PVA-NC-EP4), as shown in Figure 4. The high bending strength indicates the high degree of crosslinking between the resin and hardener [37,38]. The 10PVA-NC-EP3 sample showed the lowest flexural strength as compared to other cases. It might be due to more epoxy groups reacting with the hydroxyl groups or via homo-polymerization, which ultimately makes the samples brittle [39,40]. The sample with 4 wt% hardeners (10PVA-NC-EP4) showed the highest elongation-at-break compared to other concentrations because the carbon-amine nitrogen crosslinking instigates more flexibility in the sample, i.e., the chains in the material are stretched to a high degree before they break [37,39,40,41].

In the 15 wt% PVA cases, the bending strength increased from 45.82 to 55.41 MPa, while the hardener content increased from 3 to 4 wt%. The 4 wt% hardener case (15PVA-NC-EP4) showed the highest flexural strength compared to other cases, as illustrated in Figure 4. The flexural modulus was 3.46 and 4.25 GPa for EP3 and EP4 resin cases. The flexural modulus gradually increased by increasing the hardener content; it might be due to the epoxy ring reaction’s opening by the hydroxyl groups, leading to a more stable structure. The elongation-at-break decreased as the hardener content increased; it was negatively affected when the strength and modulus were enhanced [38].

In the cases of 20 wt% PVA-blended NC composites, as shown in Figure 4, the bending strength increased with the hardener content, but they were lower than the 15 wt% PVA cases. The flexural modulus and elongation-at-break also decreased. The 15PVA-NC-EP4 case exhibited the highest flexural strength of 55.41 MPa and modulus of 4.25 GPa. Thus, this content is optimum for 3D printing the PVA-blended NC paste.

The current study of 3D-printed nanocellulose infused Epofix flexural properties are comparable with the natural fibers reinforced epoxy composites manufactured via traditional infusion and hand-layup processes [42]. Fiore et al. [43] reported a flexural strength of jute/epoxy composites of 56.4 MPa. Similarly, Yusoff et al. [44] obtained a flexural strength of 51 MPa with oil palm fibers infused epoxy composites, and Venkateshwaran et al. [45] obtained 57.53 MPa flexural strength with banana fiber/epoxy composites.

#### 3.2.2. PVA-Blended NC Green Composites

Furthermore, green composites were printed with 10 and 15 wt% PVA-blended high-concentration NC pastes with optimized twin-screw extruder printing parameters. The printed composites were dry at cleanroom conditions (relative humidity: 45%, and temperature: 25 °C) to facilitate shape fidelity for 5 days. As confirmed by SEM, the dried composites showed a strong bilayer adhesion due to the improved interfacial crosslinking, as confirmed by FTIR and SEM. The dried composites are shown in Figure 5a.

The flexural test was conducted to analyze the effect of blending high concentrations of PVA (15 and 20 wt%) with NC on the strength of the 3D-printed composites. Figure 5b shows the printed green composites’ flexural strength and modulus. The green composites printed with 15 wt% PVA-blended NC paste exhibited a bending strength, modulus, and strain-at break of 94.78 ± 3.18 MPa, 9 ± 0.27 GPa, and 1.04, respectively. However, with 20 wt% PVA-blended NC paste, the green composites exhibited a bending strength of 74.99 ± 1.71 MPa, bending modulus of 7.81 ± 0.46 GPa, and strain-at-break of 1.01, as shown in Figure 5b. The mechanical results gradually decrease with the increase of blending wt% of PVA. This behavior may be due to self-crosslinking initiated by an undue wt% of PVA-blended with NC paste, which hindered the crosslinking network formation between nanocellulose chains [46]. Furthermore, excess PVA leads to brittleness, reducing the green composites’ strain-at-break. The current study’s mechanical results are higher than those reported in recent studies [35,36]. Hausmann et al. reported a flexural strength of green composites of 40 MPa using a wet densification process after 3D printing nanocellulose [36]. In contrast, Klar et al. [35] reported a tensile strength of 17.6 MPa of printed nanocellulose green composites. The low mechanical properties in the previously reported studies are due to low interfacial adhesions between the printed layers due to concentrated nanocellulose paste. However, the present study’s flexural strength results are 136.9% higher than the densified structures [36].

### 3.3. Structural Characterizations

FTIR spectra were carried out on 3D-printed composites to verify the interaction between constituents, as shown in Figure 6. The high-concentration NC paste exhibited a characteristic peak around 3300–3400 cm^−1^, associated with the –OH groups stretching vibrations. The absorptions band around 1642 and 1110 cm^−1^ corresponds to -OH vibrations and stretching of -C-O groups, respectively [47]. The band at 1615 cm^−1^ is attributed to the carboxylate group (-COO-) due to the TEMPO oxidation of hardwood pulp to obtain cellulose [48].

The 3D-printed 15PVA-NC depicted the characteristic bands at 3443 cm^−1^ and 1721 cm^−1^, corresponding to the –OH and –COCH_3_ groups, respectively [49,50]. After mixing 15 wt% PVA concentration in NC paste, the –OH band shifted towards the lower wavenumber due to the hydrogen bonds [49,51]. Due to the aromatic stretching, the resin-infused 15PVA-NC-EP4 depicted the characteristic peaks at 3029 cm^−1^ and 1500–1610 cm^−1^. However, the –NH stretching peak around 3257 cm^−1^ merged with the –OH peak. The band near 1721 cm^−1^ did not appear after curing the 3D-printed structure because of the –NH group and the –COCH_3_ group interaction [52]. It confirms the resin’s successful curing and interaction with the PVA-NC paste.

Freeze drying is a procedure whereby structures are frozen in a refrigerator, and then the frozen solvents are evaporated through a sublimation process under a vacuum, forming porosity within the dried structures. Therefore, the SEM images were taken for the freeze-dried structures before and after resin infusion to evaluate the structures’ porosity. The freeze-dried samples were very fragile before the resin infusion. Figure 7a shows the cross-sectional SEM image of the 3D-printed 15PVA-NC specimen after freeze-drying. The cross-section shows micro to macro-scale porous structures. This porous structure was due to the sublimation process, where ice particles were directly evaporated during the freeze-drying process. The pore size varies and can be decreased by lowering the freezing temperature [53]. The freeze-dried structures showed a proper layer-wise pattern without delamination or adhesion issues between the layers.

Furthermore, Figure 7b shows the fracture surfaces of the 15PVA-NC-EP4 specimen after the TPB test. It is clearly shown that the resin was distributed uniformly in most pores of the 3D-printed parts and between the printed layers. Almost all pores in the structure were filled with resin due to the cellulose’s hydrophilic nature. The resin was infused via the infusion process into the freeze-dried structures. Additionally, the resin showed firm interfacial adhesion with the 3D-printed layers, consequently increasing the mechanical properties.

Similarly, Figure 8a exhibited the cross-sectional surface of the green composites printed with 15 wt% PVA-blended NC paste after the TPB test. The green composites were slowly dried after printing in cleanroom conditions. The slow drying resulted in the high-shaped fidelity of the printed structures. As shown in Figure 8a, the strong bilayer adhesion can be seen clearly in the images. This strong adhesion between the printed layers is attributed to improved crosslinking via hydrogen bonding between PVA and NC hydroxyl groups. Furthermore, the interfacial adhesions at the printed composites’ sides also reflect the strong adhesion between the printed bilayers without pores, as shown in Figure 8b. The strong adhesion between the printed layers is responsible for the green composites’ mechanical performance.

## 4. Conclusions

This paper has demonstrated an additive manufacturing process for 3D-printing composites. Two different composites, i.e., Epofix-reinforced NC composites and PVA-blended NC green composites, were fabricated through 3D printing containing the highest possible concentrations of nanocellulose (CNC-CNF). For Epofix-impregnated composites, the high concentration of NC paste was 3D printed in a layer-wise pattern structure similar to electrospun fibers, followed by freeze-drying and impregnation of resin via the VARTM process. Furthermore, the resin’s hardener concentration effect was investigated on the mechanical properties of the 3D-printed structures. The 15 wt% PVA-blended NC paste structure infused with EP4 resin showed the highest bending strength of 55.41 MPa and modulus of 4.25 GPa. In contrast, the directly 3D-printed green composites with 15 wt% PVA-blended NC paste showed a bending strength and modulus of 94.78 ± 3.18 MPa and 9.003 ± 0.268 GPa, respectively. The proposed NC 3D printing technology of high-concentration nanocellulose composites could open new ways to compete with electrospinning, followed by traditional weaving methods to save time and cost without sacrificing the mechanical performance of composites.

## Figures and Tables

**Figure 1 polymers-15-00669-f001:**
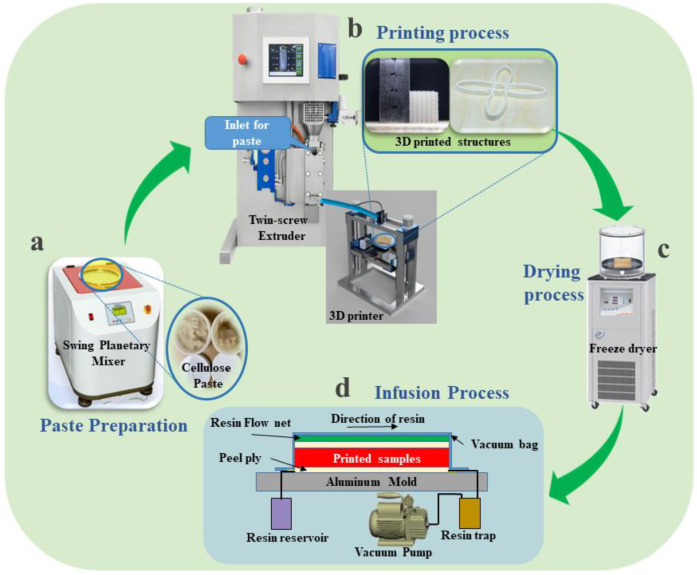
A schematic of NC 3D printing by extrusion: (**a**) paste preparation using a swing planetary mixer, (**b**) 3D printing by extruding NC paste, (**c**) freeze-drying, and (**d**) resin infusion.

**Figure 2 polymers-15-00669-f002:**
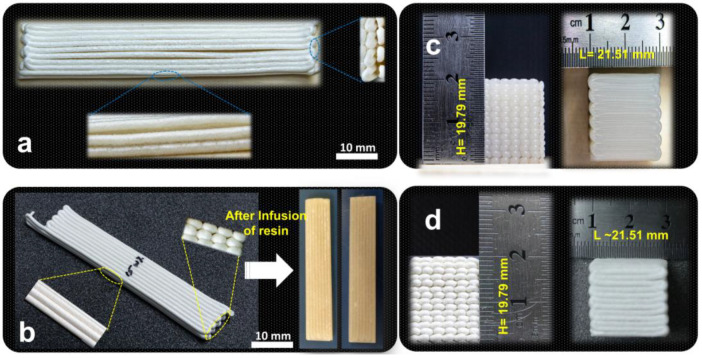
Digital images of 3D-printed and freeze-dried samples: (**a**) pure NC paste, (**b**) 15 wt% PVA-blended NC paste, (**c**) wet cubic sample after printing with 15 wt% PVA-blended NC paste, and (**d**) freeze-dried cubic sample printed with 15 wt% PVA-blended NC paste.

**Figure 3 polymers-15-00669-f003:**
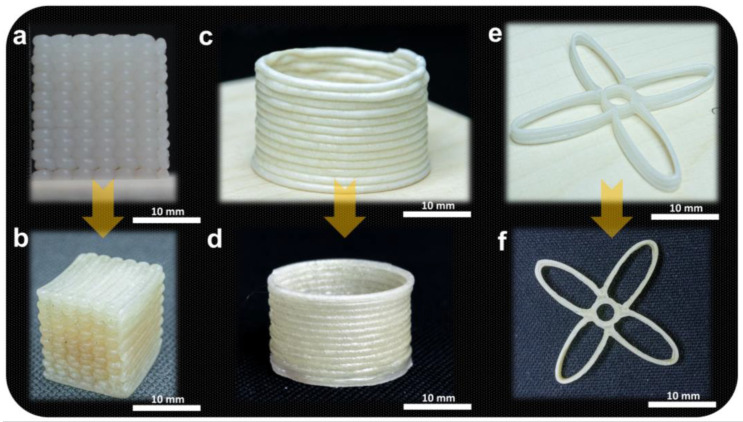
Photographs of 3D-printed, resin-infused, and dried samples: (**a**) wet cubic structure, (**b**) infused and dried cubic structure, (**c**) wet cylindrical structure, (**d**) infused and dried cylindrical structure, (**e**) wet flower-shaped structure, and (**f**) infused and dried flower-shaped structure.

**Figure 4 polymers-15-00669-f004:**
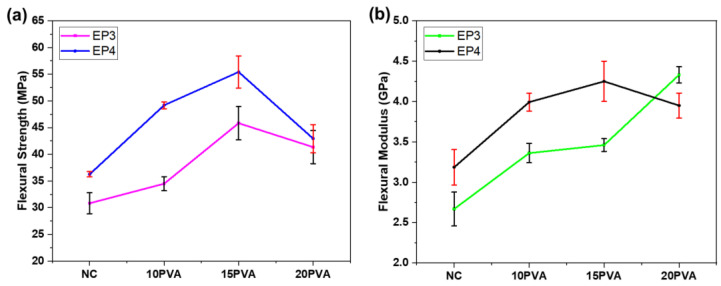
Pristine and PVA-blended NC composites (**a**) Flexural strength and (**b**) Flexural modulus.

**Figure 5 polymers-15-00669-f005:**
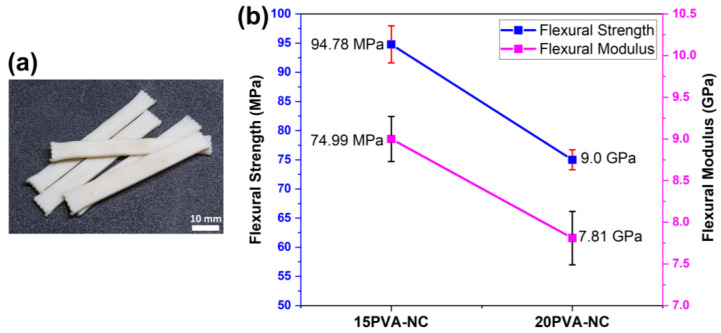
The 3D-printed PVA-blended composites (**a**) Photographs of dried composites at cleanroom conditions and (**b**) Flexural Strength and Modulus of PVA-blended NC Green composites.

**Figure 6 polymers-15-00669-f006:**
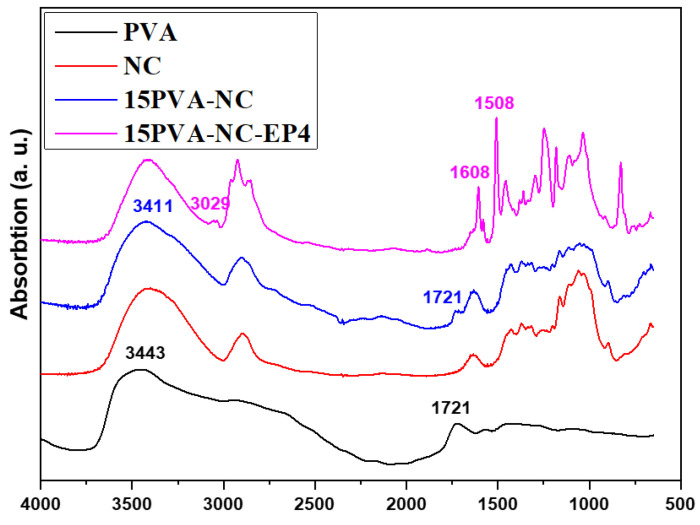
FTIR of the pristine, PVA-blended, and resin-infused NC samples.

**Figure 7 polymers-15-00669-f007:**
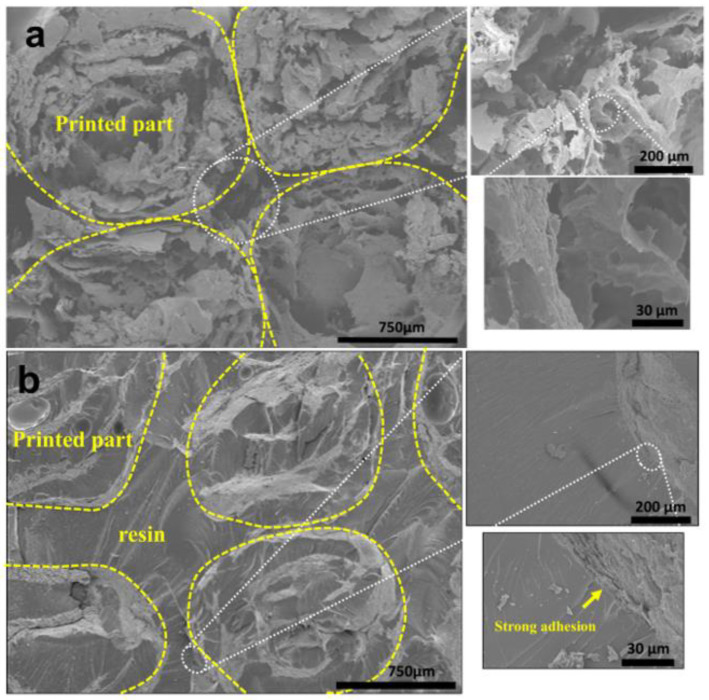
Cross-sectional SEM images: (**a**) before and (**b**) after resin infusion.

**Figure 8 polymers-15-00669-f008:**
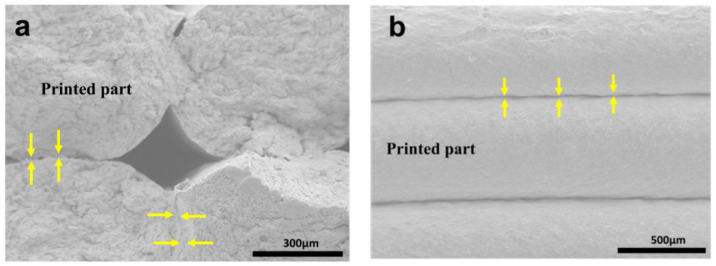
Cross-sectional SEM images of cleanroom-dried 15 wt% PVA-blended NC samples: (**a**) after TPB test, and (**b**) interfacial adhesion.

**Table 1 polymers-15-00669-t001:** Mechanical properties of 3D-printed freeze-dried infused Epofix resin samples.

Mechanical Properties	Resin	PVA Content (wt%)			
		0	10	15	20
Flexural strength (MPa)	EP3	30.82 ± 1.98	34.49 ± 1.32	45.82 ± 3.21	41.34 ± 3.09
	EP4	36.25 ± 0.49	49.18 ± 0.64	55.41 ± 3.01	42.93 ± 2.63
Flexural modulus (GPa)	EP3	2.67 ± 0.21	3.36 ± 0.12	3.46 ± 0.08	4.33 ± 0.10
	EP4	3.19 ± 0.22	3.99 ± 0.11	4.25 ± 0.25	3.95 ± 0.15
Strain-at-break (%)	EP3	1.23 ± 0.16	1.05 ± 0.07	1.48 ± 0.01	0.98 ± 0.04
	EP4	1.19 ± 0.10	1.31 ± 0.03	1.33 ± 0.06	1.14 ± 0.03

EP3: Epofix: Hardener = 25:3 (*w*/*w*); EP4: Epofix: Hardener = 25:4 (*w*/*w*).

## Data Availability

The data presented in this study are available on request from the corresponding author.

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
