# Peer review of "Additively-Manufactured High-Concentration Nanocellulose Composites: Structure and Mechanical Properties"

_polymers, 2023, doi:10.3390/polym15030669_

Round 1

Reviewer 1 Report

This article is devoted to the preparation and study of composites with a high concentration of nanocellulose. The article presents a lot of experimental data that are well and adequately described. The abundance of different methods of analysis make a good impression. However, I recommend that the following points be improved:

1. In the introduction, you can add general information about cellulose and its derivatives and their applications.

2. Please cite: 10.1007/s00226-022-01363-4.

3. It is desirable to add more references to the literature and comparisons when describing physicochemical methods.

4. Microscopic and spectroscopic data can be described more voluminously.

5. Description of the use of material for 3D printing is also desirable to expand. Please compare the obtained data with those available in the literature in more detail.

Author Response

Reviewer #1

On behalf of the authors, I would like to appreciate your valuable comments. This is the list of answers to the comments. Revised parts were highlighted in red in the manuscript.

Comments and Suggestions for Authors

This article is devoted to the preparation and study of composites with a high concentration of nanocellulose. The article presents a lot of experimental data that is well and adequately described. The abundance of different methods of analysis makes a good impression. However, I recommend that the following points be improved:

=> Answer: Thank you for your comments.

  1. In the introduction, you can add general information about cellulose, its derivatives, and its applications.

=> Answer: Thank you for your comment. The introduction section includes general information about cellulose and its applications.

  1. Please cite 10.1007/s00226-022-01363-4.

=> Answer: Thank you for your comment. We have cited that paper in the introduction section.

  1. It is desirable to add more references to the literature and comparisons when describing physicochemical methods.

=> Answer: Thank you for your comment. We have revised explanations in the manuscript and included comparison results cited in new references.

  1. Microscopic and spectroscopic data can be described more voluminously.

=> Answer: Thank you for your comment. We have revised the manuscript's microscopic and spectroscopic results, citing some new research articles.

  1. Description of the use of material for 3D printing is also desirable to expand. Please compare the obtained data with those available in the literature in more detail.

=> Answer: Thank you for your comments. We compared our 3D-printed high-concentration nanocellulose-impregnated Epofix resin mechanical results with the literature.

Reviewer 2 Report

In this article, two types of composites containing PVA and NC have been printed, and the mechanical, thermal, and microstructural properties have been investigated by three-point bending, FTIR, and SEM. The article's topic is exciting and innovative, but the different sections are poorly presented and need fundamental corrections.

The abstract should appeal to the reader. Novelty should be presented transparently. Research achievements should be mentioned quantitatively. The purpose of doing the work should also be mentioned.

In the research method section, it is suggested to summarize the printing parameters in a table.

In section 2.4, add more explanations about the infusion method. If possible, use pictures or schematics.

The introduction section needs fundamental correction, especially regarding the importance of 3D printing and new works in this field. It is suggested to use the following sources to complete this section (“3D printing of PLA-TPU with different component ratios: Fracture toughness, mechanical properties, and morphology” “Development of Pure Poly Vinyl Chloride (PVC) with Excellent 3D Printability and Macroand MicroStructural Properties” and “A New Strategy for Achieving Shape Memory Effects in 4D Printed Two-Layer Composite Structures).

Sections 2.3 to 2.6 can be merged into the research method.

Why are the three-point bending test dimensions different in the two cases? Doesn't this dimension change affect the extracted properties? What is the number of test repetitions for each sample?

Lines 162 to 176 of the results section should be presented in other sections. This paragraph is the overview of the current research and cannot be included in the results section.

Add a scale bar to all images that are samples.

What is the reason for the nonlinear effect of PVA on flexural strength and modulus? Is it possible to add elongation changes to this figure? It is better to present the stress-strain diagram as well.

Section 3.3 should be merged into 3.1. Also, sections 3.4 and 3.5 should be merged in the previous sections.

Also, the results for the green composite should be added to figure 5 or at least to table 1 for comparison of mechanical properties.

The SEM images are raw and should be corrected, especially the scale bar and the text at the bottom.

The results section needs major revision because most of its volume is the results report. In contrast, in-depth analysis and discussion should be done in each subsection.

Author Response

Reviewer #2

On behalf of the authors, I would like to appreciate your valuable comments. This is the list of answers to the comments. Revised parts were highlighted in red in the manuscript.

Comments and Suggestions for Authors

1. In this article, two types of composites containing PVA and NC have been printed, and the mechanical, thermal, and microstructural properties have been investigated by three-point bending, FTIR, and SEM. The article's topic is exciting and innovative, but the different sections are poorly presented and need fundamental corrections.

=> Answer: Thank you for your comments. We did our best to revise the manuscript by considering the comments.

2. The abstract should appeal to the reader. Novelty should be presented transparently. Research achievements should be mentioned quantitatively. The purpose of doing the work should also be mentioned.

=> Answer: Thank you for your comments. The work's purpose, novelty, and research achievements are included in the abstract.

3. In the research method section, it is suggested to summarize the printing parameters in a table.

=> Answer: Thank you for your comments. We have summarized the printing parameters in Table S2 (Supporting Information).

4. In section 2.4, add more explanations about the infusion method. If possible, use pictures or schematics.

=> Answer: Thank you for your comments. An explanation of the infusion process has been added in the experimental section. The process pictures are added to the supporting information (Figure S2).

5. The introduction section needs fundamental correction, especially regarding the importance of 3D printing and new works in this field. It is suggested to use the following sources to complete this section ("3D printing of PLA-TPU with different component ratios: Fracture toughness, mechanical properties, and morphology" "Development of Pure Poly Vinyl Chloride (PVC) with Excellent 3D Printability and Macro‐and Micro‐Structural Properties" and "A New Strategy for Achieving Shape Memory Effects in 4D Printed Two-Layer Composite Structures").

=> Answer: Thank you for your comments. We have revised the 3D printing technology explanation and cited the abovementioned articles.

6. Sections 2.3 to 2.6 can be merged into the research method.

=> Answer: Thank you for your comment. We have revised the manuscript's structure per the reviewer’s suggestions.

7. Why are the three-point bending test dimensions different in the two cases? Doesn't this dimension change affect the extracted properties? What is the number of test repetitions for each sample?

=> Answer: Thank you for your comments. The structures were processed differently, i.e., freeze-drying followed by resin-infused and direct drying at cleanroom conditions. During freeze-drying, the structures showed negligible shrinkage. However, the cleanroom-dried samples showed shrinkage due to water evaporation. That's why the samples have different dimensions. However, both structures have been tested according to ASTM D790 standards. We have tested five samples for each case. The standard deviations are shown in Figures 3, 5, and Table 1.

8. Lines 162 to 176 of the results section should be presented in other sections. This paragraph is the overview of the current research and cannot be included in the results section.

=> Answer: Thank you for your comment. We have adjusted those lines in the experimental section.

9. Add a scale bar to all images that are samples.

=> Answer: Thank you for your comment. The scale bar for all images has been added to the manuscript.

10. What is the reason for the nonlinear effect of PVA on flexural strength and modulus? Is it possible to add elongation changes to this figure? It is better to present the stress-strain diagram as well.

=> Answer: Thank you for your comments. The mechanical results gradually decrease with the increase of blending wt% of PVA. This behavior may be due to self-crosslinking initiated by an undue wt% of PVA blended with NC paste, which hindered the crosslinking network formation between nanocellulose chains [46]. Furthermore, excess PVA leads to brittleness, reducing the green composites' strain-at-break. Furthermore, we included the PVA-NC sample's stress-strain curves in the supporting information (Figure S4).

11. Section 3.3 should be merged into 3.1. Also, sections 3.4 and 3.5 should be merged in the previous sections.

=> Answer: Thank you for your comments. We have revised the manuscript's structure per the reviewer’s suggestions.

12. Also, the results for the green composite should be added to figure 5 or at least to table 1 for comparison of mechanical properties.

=> Answer: Thank you for your comments. The results have been included in Figure 5.

13. The SEM images are raw and should be corrected, especially the scale bar and the text at the bottom.

=> Answer: Thank you for your comments. The SEM images have been corrected.

14. The results section needs major revision because most of its volume is the results report. In contrast, in-depth analysis and discussion should be done in each subsection.

=> Answer: Thank you for your comments. We have compared our obtained results with the reported literature results.

Round 2

Reviewer 2 Report

Accept